# Can PPAR *γ* Keep Cadmium in Check?

**DOI:** 10.3390/biom12081094

**Published:** 2022-08-09

**Authors:** Caila Robinson, Richard F. Lockey, Narasaiah Kolliputi

**Affiliations:** Department of Internal Medicine, Division of Allergy and Immunology, Morsani College of Medicine, University of South Florida, Tampa, FL 33612, USA

**Keywords:** apoptosis, lung injury, oxidative stress, metal toxicity

## Abstract

Cd, a naturally occurring endocrine toxin found in tobacco leaves, originates in the environment and enters the body through inhalation, targeting the lungs and kidneys. A study published by Larsen-Carey et al. revealed that cadmium mediates the persistence of classically activated lung macrophages to exacerbate lung injury. The research discovered a novel role for PPAR *γ* as an effective regulator for the alternative activation of macrophages in response to Cd and Cd-induced lung injury.

Over 30 million people in the U.S. smoke cigarettes [1]. Most of them are aware that smoking poses severe health risks, but very few understand that toxic heavy metals, such as Cadmium (Cd), accumulate in the tissues. Cd, a naturally occurring endocrine toxin found in tobacco leaves, originates in the environment, and enters the body through inhalation, targeting the lungs and kidneys [2]. When comparing smoking status, smokers versus nonsmokers, Cd bioaccumulation differs between the two. Among cigarette smokers, Cd is transferred as mainstream smoke at a relatively higher concentration, reflecting how tobacco products represent the greatest environmental source of Cd exposure via inhalation. Among nonsmokers, the primary source of exposure is oral exposure through the diet [3]. High levels of Cd induce various system failures in the body, including the respiratory system. Cd, as a result, is a recognized public health concern due to elevated cigarette smoking, especially seen in underrepresented and low-income communities. However, it is unclear whether the damage to lung tissue caused by Cd is attributed to inflammation of activated macrophages. Furthermore, little is known about the inflammatory response and impact of M1 macrophage polarization upon Cd exposure [2]. M1 and M2 macrophages are an essential immune response for the emergence of fibrosis. Thus, the contribution of Cd toward lung disease and fibrosis, among others, is a widely researched area.

Lung macrophages behave as a layer of protection and have a crucial role in the inflammatory response and injury repair [4]. Different macrophage phenotypes function in response to oxidative stress [2]. TNFα activates M1 macrophages while IL-10 generally activates M2 macrophages [1]. When activated by Cd, macrophages alter the metabolic functions of glycolytic flux [5]. Macrophage dysfunction is elevated by Cd [6]. However, when the macrophages’ polarization is disrupted by exposure to Cd, their responsibility shifts [2]. Cd exposure results in an inflammatory response through activation of M1 macrophage or conversely, triggers an anti-inflammatory pathway through activation of M2 macrophages [7]. The switch from the M1 to the M2 macrophage response seems to advance with increasing dose or chronic exposure to smoke opposed to low dose; however, the M2 response may also be suppressed from the loss of the T cell function and suppression of IL-4 and IL-5 production [8]. Low doses or concentrations often induce activation of the M1 inflammatory response, up to a certain threshold, after which the M1 response is suppressed while the M2 polarization becomes enhanced. Smoking results in a systemic inflammatory response in which the high dose exposure of Cd inhibits the required T cell response [8]. Therefore, the seemingly paradoxical effect can be explained as such: greater pro-inflammatory cytokine activity leads to M1 macrophage activation while a greater phagocytic activity contributes to the differentiation to M2 macrophage activation [8]. As a result, the M1/M2 lung macrophages regulation would be a beneficial target of therapeutic agents to mediate lung injury [9]. Accordingly, cellular metabolism contributes to macrophage polarization [4] to mediate the secretion of inflammatory cytokines [7]. Macrophage reprogramming requires knowledge of the mechanisms that are promoted, but this field is not well understood.

Larsen-Carey et al. exposed wildtype (WT) mice to CdCl_2_ and measured the expression to understand the implications of Cd accumulation in the lungs [4]. The lung macrophages were treated with Cd and analysis of TNFα, iNOS, arginase 1, TGF-β1, IL-10, PDFG-B and the positive control, LPS, was performed [4]. Larsen-Casey et al. compared both exposed and isolated macrophages to demonstrate the impact of Cd. This study was performed to capture the immune response since Cd operates as an immune suppressant [2]. The treatments revealed how Cd treated macrophages develop mechanisms for metabolic reprogramming to the pro-inflammatory phenotype in response to injury [4]. It was deduced that Cd accumulation regulated phenotypic switching. In the study, there was greater Cd burden in the lung tissues due to suppressed inflammation likely caused by higher Cd concentrations from chronic smoking. Contrarily, Cd at low doses with pro-inflammatory genes can increase immune cells through genetic and metabolic responses which can be seen in H1N1 infections [10]. Together these processes result in decreased immune cell counts and cytokine levels [10].

Mice exposed to CdCl_2_ exhibited reduced expression of mRNA in arginase-1, TGF-β1, IL-10, and PDGF-B [4], and an increase in HIF-1α relative to controls. Hence, increased HIF-1α expression allows for metabolic reprogramming of macrophages [4]. This suggests that persistent Cd exposure disturbs macrophage activity through an activated inflammatory response, resulting in lung injury or disease. These findings also indicate that PPAR *γ* can be used to decrease Cd activated macrophages when overexpressed [5]. PPAR *γ* controls gene expression by acting as an inflammatory regulator. Accordingly, the Cd-induced macrophages revealed elevated levels of glyceraldehyde 3-phosphate with PPAR *γ* expression [4]. Larsen-Casey et al. noted the occurrence of glycolytic changes in response to Cd exposed macrophages while metabolic reprogramming in PPAR *γ* treated macrophages fail toward glycolysis [4].

Cd is considered a lethal metal found in cigarettes and is a recognized carcinogen. The amount of Cd in a cigarette ranges between 1 and 2 μg and is known to collect in the lungs [5]. Cigarette smoking contributes to lung diseases such as chronic obstructive pulmonary disease (COPD) and cancer [1]. To improve human health, including that of those who are casual smokers or addicted, future research can develop ways to make cigarette use safer while simultaneously treating the damage caused by Cd inhalation. Will Cd continue to dominate as a powerful toxin in cigarettes? Larsen-Casey et al. noted that PPAR *γ* is an anti-inflammatory regulator in macrophages [4], thus revealing the impact of Cd in cigarettes versus electronic cigarettes on the lungs when inhaled. The deleterious effects of aerosol and smoke inhalation would also be determined. Although incorporating PPAR γ into cigarettes would prove to be beneficial in reducing Cd toxicity on the lungs, it would not be transported into its active form in the tobacco smoke. Novel research is required to direct PPAR *γ* to the affected cells damaged by Cd. Additionally, for those already suffering from a Cd-induced lung injury, investigation into altering PPAR *γ* as a therapeutic mechanism to minimize and manage injury should be explored.

Although the findings provided by Larsen-Casey et al. suggest that PPAR *γ* is an effective regulator for the alternative activation of macrophages in response to Cd [4], Cd-induced lung injuries will continue to be a detriment to human health. This is attributed to macrophage activation following Cd exposure [4]. According to Larsen-Casey et al., synthetic and natural ligands activate the PPAR *γ* inflammatory regulator [4], yet Cd will negatively impact the cellular and innate immune functions [5]. The resulting natural products can act as PPAR *γ* agonists and antagonists [11]. Post-translational modification (PTM) is a mechanism known to regulate PPAR *γ* for several metabolic diseases relating to lung injury [11]. Activated PPAR *γ* can therefore regulate HO-1 (heme oxygenase-1) which in turn impacts injury due to inflammation, reactive oxygen species (ROS) production and apoptosis [11]. These PTMs include phosphorylation, SUMOylation, ubiquitination, acetylation, and glycosylation, in which most of the mechanisms are involved in glucose metabolism [11]. The use of PPAR *γ* for biological processes is critical for cell differentiation, survival, and apoptosis; it also strongly contributes to the regulation of glucose metabolism. This indicates that glucose use and glycolysis is greatly enhanced in the presence of PPAR *γ* [12]. As a critical yet novel transcription factor, PPAR *γ* can switch on particular isoenzymes to promote both glycolysis and cell proliferation [12]. There was no proposed mechanism by Larsen-Casey et al. for lungs to control the imbalance between the phenotypic switching for regulation of the inflammatory response caused by Cd exposure. For Cd accumulation the question remains: will PPAR *γ* activation only provide a transient solution to the suppressed inflammatory response from Cd-induced oxidative stress?

In summary, PPAR *γ* may prove successful in containing the effects of lung injury caused by Cd. M1 lung macrophages are polarized in response to oxidative stress [9], making them crucial to understanding the mechanisms underlying macrophage reprogramming. Macrophage polarization remains essential to naturally restore the body after extensive inflammatory injury [13]. Emergence of a pro-inflammatory macrophage phenotype will continue to delay lung injury recovery without understanding the implications from the level of Cd exposure and its effects on innate immunity. Cd regulates macrophage polarization and represents a proportional relationship between the degree of exposure and the development of lung injury or disease [4]. Although PPAR *γ* is recognized as an effective agent in insulin resistance medication for type II diabetes [14], there is little known about how PPAR *γ* can manage Cd-mediated lung injuries. Therefore, more research is required to provide a sound solution using PPAR *γ* to precisely target Cd-induced injury.

## Data Availability

Not applicable.

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
