# Peer review of "Can PPAR γ Keep Cadmium in Check?"

_biomolecules, 2022, doi:10.3390/biom12081094_

Round 1

Reviewer 1 Report

In this work the authors provide a new perspective about the treatment of Cadmium-induced lung injury involving PPAR gamma as a critical regulator of the anti-inflammatory phenotype in macrophages. Indeed, a previous study by Larsen-Carey et al. revealed that Cadmium mediates the persistence of classically activated macrophages to exacerbate lung injury, and PPAR gamma could play a novel role as a regulator for the alternative activation of macrophages through metabolic reprogramming. Being PPARs activated by both synthetic and natural ligands, these may be a novel target for regulating macrophage metabolic reprogramming and influencing lung repair caused by Cadmium exposure.

However, besides the activation by ligands, which is often accompanied by serious side effects related to their agonism, PPAR gamma is also regulated by several post-translational modifications (PTM) (1), an aspect which is not mentioned in this work.  According to literature, those mechanisms seem to be of very importance in inflammatory cells and cells involved in glucose metabolism (2).  

In my opinion, in the search of a new 'perspective', this aspect cannot be ignored and should be added to this work. 

(1) Carvalho et al. PPAR gamma: from definition to molecular targets and therapy of lung diseases. Int. J. Mol Sci. 2021, 22(2):805.

(2) Shu et al. Phosphorylation of PPAR gamma at Ser84 promotes glycolisis and cell proliferation in hepatocellular carcinoma by targeting PFKFB4. Oncotarget. 2016, 7(47):76984-76994 

Reviewer 2 Report

The perspective article consists of a limited analysis of some published data on macrophage polarization and pulmonary pathogenesis as a consequence of inhalation exposure to cadmium. The manuscript requires revision in order to render it acceptable for publication.

1. "Humans are also exposed to Cd in their diet; however, inhalation is the most common (3)." The sentence is poorly worded. The most common what? If the intent was to state that inhalation is the most common route of exposure, this would be incorrect. Among cigarette SMOKERS, cadmium is the major source of cadmium exposure, and that exposure occurs by inhalation. Among nonsmokers, the major route of exposure is oral exposure through the diet. This statement needs to be reworded and corrected. It might be necessary to add a citation in order to correct the statement.

2. The authors make an effort to support the statement describing the response of macrophages to cadmium, "However, when the macrophages’ polarization is disrupted by exposure to Cd, their responsibility shifts (2)", but they do not address the apparent contradiction in the statement, "Cd exposure results in an inflammatory response through activation of M1 macrophage or conversely, triggers an anti-inflammatory pathway through activation of M2 macrophages (6)." In chapter 5, page 135 of Cigarette Smoke and Oxidative Stress (B.B. Halliwell and H.E. Poulsen, eds., Springer, Germany, 2006) I. Rahman (chapter author) stated a similar apparent contradiction. It was stated that various "...constituents of cigarette smoke, such as, nicotine, acrolein, hydroquinone, catechol, and 4-HNE, inhibits either basal or LPS-induced NF-κB activation and the activation of NF-κB-dependent genes such as IL-1, IL-2, IF-γ, TNF-α, and IL-8, in the U-937  cell line or peripheral blood-derived monocytes (Ouyang et al., 2007, Sugono et al., 1997). However, in the next paragraph, the opposite, increased expression of NF-κB, is described. The apparent paradox described by the authors of the perspective article under consideration (proinflammatory versus suppression of inflammatory response) and that described by Rahman (suppression versus activation of NF-κB activity) are not necessarily contradictions, and they should be explained to the reader in order to avoid confusion with regard to the apparent contradiction. The authors should read pages 1189, 1191, 1192 in "Pappas RS. Toxic elements in tobacco and in cigarette smoke: inflammation and sensitization. Metallomics 2011;3:1181-1198. The topic of T Helper cell and Macrophage polarization is discussed in terms of dose-dependence in exposures to cigarette smoke, which of course includes cadmium in particulate. The authors should consider that low doses induce activation of M1 inflammatory response, up to a threshold, after which, M1 response is suppressed and M2 polarization is enhanced. They do not have to adopt this view, but they should at least consider it in explaining the apparent contradiction between proinflammatory versus suppression of inflammatory responses in macrophages in two different statements. An explanation such as this eliminates the apparent contradiction.

3. Bottom of first page, continuing on second page: "Larsen-Casey et al. compared both exposed and isolated macrophages to demonstrate the impact of Cd. This study was performed to mediate the immune response since Cd operates as an immune suppressant (2)." The word mediate does not belong here, and should be replaced with a better choice of word. However, the statement also presents only one side of the story of the immunotoxicology of cadmium inhalation. The authors should read, "Chandler JD, Hu X, Ko EJ, Park S, Fernandes J, Lee YT, Orr ML, Hao L, Smith MR, Neujahr DC, Uppal K, Kang SM, Jones DP, Go YM. Low-dose cadmium potentiates lung inflammatory response to 2009 pandemic H1N1 influenza virus in mice. Environ Int. 2019 Jun;127:720-729. doi: 10.1016/j.envint.2019.03.054." At low doses, cadmium increases inflammatory response, whereas the authors' citation of Larsen-Casey et al. only describes the suppression of inflammation, likely due to a higher cadmium exposure, such as occurs among chronic smokers.

4. The authors state, "Larsen-Casey et al. noted that PPAR ? is an antiinflammatory regulator in macrophages (4) thus, can be used as a coating in cigarettes to reduce the detrimental effects of Cd on the lung when inhaled." This is a poorly thought out suggestion. PPAR ? is a protein. Proteins do not respond well to high temperatures and combustion. At minimum, heat denatures the proteins, at worst, they are decomposed. PPAR ? would not be transported in active form in tobacco smoke.

Author Response

Please see atcched file

Round 2

Reviewer 1 Report

The author provided a revised version of the manuscript, including corrections according to the reviewers' comments. In the present form, the paper is more exhaustive and can be considered as suitable for publication

Author Response

The author provided a revised version of the manuscript, including corrections according to the reviewers' comments. In the present form, the paper is more exhaustive and can be considered as suitable for publication

We would like to thank Reviewer 1 for their time and extensive review to asses the manuscript.

Reviewer 2 Report

The manuscript is in much better shape after the first revision. However, the conclusions made on page 3 (last two paragraphs) show incomplete understanding of macrophage polarization and regulation of proinflammatory status. With increasing exposure to cigarette smoke particulate in general, and cadmium in the particulate, Macrophages are already naturally repolarized toward the M2 phenotype. The M1/M2 polarization status is regulated by the body. The authors state, "There was no therapy proposed by Larsen-Casey for lung repair using a macrophage reprogramming mechanism to regulate the inflammatory response caused by Cd exposure". The authors then propose that further therapeutic  intervention is needed in order to enhance the phenotype conversion from M1 to M2. This is not a good idea at all. Smokers are negatively impacting pulmonary health by inhaling cadmium and other toxicants from tobacco smoke. Depending on the level of exposure, the lungs have already shifted polarization in the direction of the M2 phenotype. If the shift did not occur at all, M1 phenotype would more rapidly be induced to cause pulmonary damage due to inflammation. However, because of this pathological suppression of the M1 phenotype, smokers are more susceptible to intracellular infections from communicable disease. Smoking increases the risk for severe outcomes and hospitalization as consequences of covid19, influenza, and bacterial infections because of suppression of the M1 phenotype by smoking. An attempt to avoid even more inflammatory activity by using a "therapy" to further polarization would not be therapeutic, but would result in reduced innate immunological resistance to pulmonary pathogens. The pulmonary damage would simply have a different cause. Therefore, the authors' conclusions need to be reconsidered.
